# Deep mutational scanning of *Pneumocystis jirovecii* dihydrofolate reductase reveals allosteric mechanism of resistance to an antifolate

Francois D. Rouleau [1,2,3,4]*, Alexandre K. Dubé [1,2,3,5], Isabelle Gagnon-Arsenault [1,2,3,5], Soham Dibyachintan [1,2,3,4], Alicia Pageau [1,2,3,4], Philippe C. Després [1,2,3,4], Patrick Lagüe [1,2,3,4], Christian R. Landry [1,2,3,4,5]

1 Institut de Biologie Intégrative et des Systèmes (IBIS), Université Laval, Québec, Québec, Canada, 2 Département de Biochimie, de Microbiologie et de Bio-Informatique, Faculté des Sciences et de Génie, Université Laval, Québec, Québec, Canada, 3 Regroupement Québécois de recherche sur la fonction, la structure et l'ingénierie des protéines (PROTEO), Université du Québec à Montréal, Montréal, Québec, Canada, 4 Centre de recherche en données massives de l'Université Laval (CRDM_UL), Québec, Québec, Canada, 5 Département de Biologie, Faculté des Sciences et de Génie, Université Laval, Québec, Québec, Canada

* Francois.rouleau.2@ulaval.ca

**Data Availability Statement:** Raw read sequencing from competition assay is available at BioProject accession: https://www.ncbi.nlm.nih.gov/

## Abstract

*Pneumocystis jirovecii* is a fungal pathogen that causes pneumocystis pneumonia, a disease that mainly affects immunocompromised individuals. This fungus has historically been hard to study because of our inability to grow it *in vitro*. One of the main drug targets in *P. jirovecii* is its dihydrofolate reductase (PjDHFR). Here, by using functional complementation of the baker's yeast ortholog, we show that PjDHFR can be inhibited by the antifolate methotrexate in a dose-dependent manner. Using deep mutational scanning of PjDHFR, we identify mutations conferring resistance to methotrexate. Thirty-one sites spanning the protein have at least one mutation that leads to resistance, for a total of 355 high-confidence resistance mutations. Most resistance-inducing mutations are found inside the active site, and many are structurally equivalent to mutations known to lead to resistance to different antifolates in other organisms. Some sites show specific resistance mutations, where only a single substitution confers resistance, whereas others are more permissive, as several substitutions at these sites confer resistance. Surprisingly, one of the permissive sites (F199) is without direct contact to either ligand or cofactor, suggesting that it acts through an allosteric mechanism. Modeling changes in binding energy between F199 mutants and drug shows that most mutations destabilize interactions between the protein and the drug. This evidence points towards a more important role of this position in resistance than previously estimated and highlights potential unknown allosteric mechanisms of resistance to antifolate in DHFRs. Our results offer unprecedented resources for the interpretation of mutation effects in the main drug target of an uncultivable fungal pathogen.

bioproject/PRJNA1013557. Demultiplexed and analyzed data with all computed scores is available in codon form in S1 File for DMSO, S2 File for IC75 and S3 File for IC90. All scripts for figures and analysis, as well as processed data, is available on GitHub at https://github.com/Landrylab/Rouleau_et_al_2024.git.

**Funding:** The project was funded by a Canadian Institutes of Health Research Foundation to CRL (387697, https://cihr-irsc.gc.ca/), a FRQNT Team Grant (2022-PR-298169, https://frq.gouv.qc.ca/en/) and a Genome Canada and Genome Quebec grant (6569, https://genomecanada.ca/). CRL holds the Canada Research Chair in Cellular Systems and Synthetic Biology (https://www.chairs-chaires.gc.ca/home-accueil-eng.aspx). FDR was supported by fellowships from FRQNT, PROTEO, EvoFunPath NSERC CREATE program and the Vanier Canada Graduate Scholarship agency. SD was supported by a fellowship from FRQNT. PCD was supported by fellowships from the FRQS and from the Vanier Canada Graduate Scholarship. The funders had no role in study design, data collection and analysis, decision to publish, or preparation of the manuscript.

**Competing interests:** The authors have declared that no competing interests exist.

## Author summary

The study of uncultivable microorganisms has always been a challenge. Such is the case of the human-specific pathogen *Pneumocystis jirovecii*, the causative agent of pneumocystis pneumonia. *P. jirovecii* is insensitive to classical antifungal drugs, making options for treatment and prophylaxis limited. In recent years, more and more cases of *P. jirovecii* infections have become resistant to treatment, highlighting the need to study and understand this pathogen's mechanisms of resistance. Here, we use a yeast strain expressing *P. jirovecii*'s DHFR as a reporter for resistance to an antifolate, one of the drug families used to treat infections. We observed that this DHFR was sensitive to methotrexate, a powerful antifolate, in a quantitative manner. Then, by using a large-scale mutational assay, we identified virtually all single mutations that confer this protein resistance to methotrexate. While many of them have also been reported in other eukaryotes, we find new mutations at positions of the protein not previously known to confer resistance or to be in contact with this competitive inhibitor. Overall, our results are a comprehensive portrait of this DHFR's resistance to methotrexate.

## Introduction

Some essential proteins are highly conserved across species, and therefore, they serve as targets of broad-spectrum drugs [1]. One such enzyme is the dihydrofolate reductase (DHFR), which catalyzes the reduction of 7,8-dihydrofolate (DHF) to 5,6,7,8-tetrahydrofolate (THF) in an NADPH-dependent manner [2]. THF is an essential cofactor involved in nucleotide and amino acid biosynthesis. Because of these properties, DHFRs are among the best-studied proteins, with several hundreds of orthologs and variants sequenced, characterized, and crystalized [3–10]. Despite their amino acid sequence diversity across orthologs, the core function and structure of DHFRs are highly conserved, and functional replacement is possible between several organisms [11,12].

Because of their essential nature, DHFRs have long been valuable targets for drug development [13]. For instance, in *Escherichia coli*, in *Plasmodium falciparum* and in humans, these enzymes have been thoroughly studied as therapeutic targets using molecules generally referred to as antifolates. Some drugs appear to inhibit many orthologous DHFRs (e.g. methotrexate (MTX) and pemetrexed) while others appear to be more specific (e.g. cycloguanil and pyrimethamine) [14,15]. Antifolates have been broadly used to treat various infections and diseases, from bacterial and fungal infections to rheumatoid arthritis and some cancers [10,12,14,16].

For many orthologs, mutations that confer resistance to antifolates have been identified. Resistance mutations act through different mechanisms depending on the specific protein and drug involved [17,18]. Indeed, antifolates have a wide range of structures, with molecules having different binding affinities and forming different contacts with the enzyme. Recently, DHFRs have been identified as potential targets for the treatment of fungal infections, where treatment options are often limited given the low number of antifungal classes [19]. Despite this, fungal DHFRs have rarely been the focus of systematic studies and therefore, their ability to evolve towards antifolate resistance remains to be elucidated. While resistance to antifolates has been rigorously investigated for some organisms of clinical interest, especially in bacteria and intracellular parasites, fungal resistance has been much less studied, even in pathogens against which antifolates have historically been administered [14].

Such is the case of *Pneumocystis jirovecii*, an opportunistic fungal pathogen and the etiological agent of pneumocystis pneumonia in humans. *P. jirovecii* was originally classified as a protozoan and named *Pneumocystis carinii*, but was officially reclassified as a fungus in 1988 with the advent of sequence-based identification [20], and renamed *P. jirovecii* in 1999 to avoid confusion with the rat-specific *P. carinii*, another member of this genus [21–23]. Members of the *Pneumocystis* genus are host-specific, and it has been reported that even amongst primates, transmission of the *Pneumocystis* associated with one species to another is very unlikely, and has not been reported as of yet [24].

*P. jirovecii* infection is most often observed in HIV patients. It is one of the most prevalent opportunistic pathogens in this population, exhibiting a high organismal load and a mortality rate of up to 20% [25–28]. In recent years, its occurrence in immunocompromised non-HIV patients, such as organ transplant recipients, has been increasing. In this group, the disease caused by *P. jirovecii* has a mortality rate close to 60% despite the use of prophylaxis. This increase raises major concerns about the future of *Pneumocystis* pneumonia in vulnerable populations [29–31]. Furthermore, *P. jirovecii* has been detected in healthy, non-immunocompromised groups, and in up to 50% of the general population, implying a higher prevalence than previously estimated [26,32–35]. The potential for this fungus to cause human infections is therefore likely to be underestimated compared to other pathogenic fungi, and warrants the development of tools for its monitoring, as well as for functional analyses.

Furthermore, *P. jirovecii* appears to be insensitive to currently used antifungal drugs such as azoles, polyenes, allylamines and echinocandins [36]. Trimethoprim-sulfamethoxazole (TMP-SMX, Bactrim) is the main drug combination used for both prophylaxis and treatment of *Pneumocystis* pneumonia [37]. In recent years, many clinical cases of *Pneumocystis* pneumonia have shown resistance to treatment, with many reporting mutations in the dihydropteroate synthase (DHPS), the target of SMX, and more recently, in DHFR, the target of TMP [38–41]. With other classes of DHPS inhibitors having acute side effects [42,43], and the toxicity associated with high doses of TMP [44], there is an urgent need to investigate the mechanisms through which antifolate resistance evolves in this fungal pathogen.

As there are currently no available methods to culture *P. jirovecii in vitro* or in animal models, let alone edit its genome, the systematic study of this fungal pathogen and of its drug resistance mechanisms remains a challenge. Here, we develop an experimental system and a resource for the study of resistance mutations to antifolates in the DHFR of *P. jirovecii* (PjDHFR). We constructed yeast strains and plasmids that allow for the complementation of the *Saccharomyces cerevisiae* DHFR with PjDHFR. We then built a library of mutants of the entire protein to comprehensively identify mutations that confer resistance to MTX, an antifolate with well-known resistance mutations and mechanisms in other eukaryotic DHFRs, including yeast, mouse, and human. We chose MTX as our antifolate of choice for this screening as *S. cerevisiae* has been previously reported as being insensitive to TMP through mechanisms not directly involving the DHFR [45]. A recent study has associated the exposure of MTX in bacteria to both the selection for, and *de novo* evolution of, TMP resistance in bacteria [46]. Additionally, MTX is often used to treat auto-immune diseases, and can act as an immunosuppressant [47], and has been associated with increased rates of *P. jirovecii* infection [48–53]. There is therefore a need to understand how MTX affects *P. jirovecii's* DHFR, and the link to resistance between these two antifolates. Finally, studying MTX resistance in the context of PjDHFR provides a tractable model for the study of underlying mechanisms of resistance-causing mutations in this protein. We use the library of mutants to conduct pooled competition assays in the presence of two different concentrations of MTX, as well as in the absence of the drug. We focus on the effects of mutations on MTX resistance. To ensure that we limit selection on mutations affecting protein function, we conduct the screening in a strain with a

functional genomic DHFR, and then reconstruct some mutants in a strain without a genomic DHFR to ensure the validity of our results. We identify sites along the protein where resistance mutations are the most prevalent and show that many mutations associated with MTX resistance in PjDHFR are found within the active site. We show that some are structurally conserved with MTX resistance mutations reported in orthologous eukaryotic DHFRs, and to other antifolates in *E. coli*. We also identify MTX resistance-conferring mutations at a position previously unreported in eukaryotes, position F199. F199 is not in direct contact with either drug, substrate, or cofactor, revealing that resistance to antifolates can evolve through allosteric mechanisms. Finally, we investigate mutations at this position to assess their impact on protein stability and binding energy with ligands using modeling. We show that mutations at F199 destabilize interaction between PjDHFR and MTX more than with DHF, and that destabilization of interaction with MTX correlates with resistance to this drug. These results imply a more complex landscape of resistance to MTX than previously suspected in eukaryotic DHFRs.

## Methods

### Strains, plasmids, and culture media

All strains used in this study were constructed from the reference strain BY4741 as an initial parent strain [54]. All transformations were conducted using the standard lithium acetate transformation protocol [55]. All PCR reactions were conducted using KAPA HiFi Hotstart DNA polymerase (Sigma-Aldrich, catalog #BK1000) unless stated otherwise. Details for all strains are available in S1 Table, plasmids in S2 Table, oligonucleotides in S3 Table, media in S4 Table and PCR reactions in S5 Table.

To test if PjDHFR is functional in *S. cerevisiae*, we had to inactivate or delete the yeast's endogenous DHFR, *DFR1*. Since it is an essential gene, to be able to grow a *dfr1* deletion strain, we had to first insert another DHFR (*dfrB1*) which would be expressed under the control of a β-estradiol induced promoter (*GAL1pr*) [75]. *GAL1pr* allows for the expression of the downstream gene in the presence of β-estradiol (Sigma-Aldrich, catalog #E2758-5G). Construction of strain IGA130 (*his3 ura3 met15 LEU2::GEM, GAL1pr::dfrB1-NATMX*) was done by inserting the *dfrB1* gene (previously known as R67—yeast codon optimized sequence) at the *GAL1* locus in strain AKD0679 (*LEU2::GEM, GAL1pr::Stuffer-NATMX*) by CRISPR-Cas9-mediated integration following the protocol detailed in [56] and using pCas plasmid with a gRNA sequence targeting the Stuffer from [57]. Transformants were selected on YPD + Geneticin 200 μg/mL (G418) + Nourseothricin 100 μg/mL (NAT). Following the loss of pCAS plasmid, the insertion of *dfrB1* in the genome was verified by colony PCR using oligos OP151-E5 and ADHterm_R with BioShop Taq DNA Polymerase and validated by Sanger sequencing. Construction of strain FDR0001 (*his3 ura3 met15 LEU2::GEM, GAL1pr::dfrB1-NATMX dfr1Δ::Stuffer*) was done using a similar CRISPR-Cas9-mediated protocol as described above, but using a Stuffer sequence and a gRNA targeting *DFR1*. Oligos DFR1_KO_F and DFR1_KO_R were used to amplify the Stuffer sequence used to knockout *DFR1*. Transformants were selected on YPD + G418 + NAT + 100 nM β-estradiol. Following the loss of pCAS plasmid, deletions were confirmed by PCR using oligos DFR1_KO_valid_F and DFR1_KO_valid_R and Sanger sequencing (BioShop Taq DNA Polymerase). Detailed PCR protocols are available in S5 Table. Details for media composition are available in S4 Table.

All yeast experiments were conducted at 30˚C for 48 h unless stated otherwise. All experiments involving bacteria were conducted at 37˚C using 2YT with appropriate antibiotics unless stated otherwise. All liquid cultures were incubated with 250 rpm orbital agitation. All Sanger sequencing was performed by the Sequencing Platform of the Centre Hospitalier de

l'Université Laval (Université Laval, Québec, Canada), and all next-generation sequencing was performed using the MiSeq Reagent Kit v3 on an Illumina MiSeq for 600 cycles at the Sequencing Platform of the Institut de Biologie Intégrative et des Systèmes (Université Laval, Québec, Canada).

Plasmid pAG416-GPD-PjDHFR was constructed from pAG416-GPD-ccdB and *Pneumocystis jirovecii* DHFR (PjDHFR) (UniPROT: A0EPZ9), codon-optimized using GenScript Biotech's GenSmart™ Codon Optimization tool, and ordered from Twist Bioscience (San Francisco, California, USA) as a Gene Fragment. pAG416-GPD-ccdB was digested using XbaI and HindIII-HF in CutSmart buffer (New England Biolabs, catalog #R0145 and #R0104), according to manufacturer's instructions, as the backbone. PjDHFR was amplified by PCR using oligos Pj_to_pAG_F and Pj_to_pAG_R under parameters specified in S5 Table, as the insert. Backbone and insert were assembled using an in-house Gibson assembly mix (1:2 vector:insert ratio) and cloned into *E. coli* MC1061 ([araD139]B/r Δ(araA-leu)7697 ΔlacX74 galK16 galE15(GalS) λ-e14-mcrA0 relA1 rpsL150(strR) spoT1 mcrB1 hsdR2) using a standard heat shock transformation method and plated onto ampicillin-containing 2YT media (100 μg/mL). Clones were confirmed by colony PCR using oligos pAG_insert_F and pAG_insert_R, and the amplicon sequence was confirmed by Sanger sequencing. The same protocol was used to construct pAG416-GPD-ScDHFR (*DFR1* insert amplified from BY4741 genomic DNA using oligos Sc_to_pAG_F and Sc_to_pAG_R) and pAG416-GPD-mDHFR (mDHFR, synthetic, L22F and F31S, insert amplified from pGEST-linker-fulllength-DHFR-HYG_B plasmid using oligos Murine_to_pAG_F and Murine_to_pAG_R).

### Functional complementation assay

The complementation assay was conducted by transforming the DHFR-containing plasmids into strain FDR0001 and plating transformations on solid SC-URA+MSG+100 nM β-estradiol media to ensure growth of transformants even in the absence of complementation. Four individual colonies were harvested from each transformation and streaked on solid SC-URA +MSG+100 nM β-estradiol media, and colony PCR was conducted to confirm transformation success. For both spot-dilution assays, transformants were grown in liquid SC-URA+MSG +100 nM β-estradiol media overnight. Cultures were diluted to 0.1 $OD_{600}$ in the same media and grown to 0.3 $OD_{600}$ to ensure that cells were in exponential phase. 1.0 $OD_{600}$ unit worth of cells were harvested from the cultures, washed, and resuspended in $ddH_2O$, and 5 μL was spotted on MTX media (See S4 Table for detailed recipe) with and without 440 μM methotrexate (BioShop, catalog #MTX440) and with and without 100 nM β-estradiol. The following spots were done by five-fold serial dilution on the same media.

### Dose response curves

Dose response curves were measured in triplicate using a Tecan Infinite M Nano (Tecan Life Sciences) plate reader. DHFR-containing plasmids were transformed into strain IGA130. Three individual colonies were sampled from transformation plates and grown individually in 3 mL SC-URA+MSG in V-bottom 24-well deep well plates overnight. The next morning, 1.0 $OD_{600}$ unit of cells was harvested and spun down at ~500 rcf for 4 minutes. The supernatant was removed, and the pellet washed with sterile $ddH_2O$, and then resuspended in 1 mL of sterile $ddH_2O$. 20 μL of this resuspension was added in a flat bottom Greiner 96 well plate, to 180 μL of liquid MTX media prepared the same day, for a final $OD_{600}$ of 0.1. Liquid MTX media was prepared using a ten-fold dilution of MTX media at 440 μM MTX to 44 μM, and then subsequent two-fold dilutions to attain 0.086 μM of MTX. Plates' edges were sealed with Parafilm to limit media evaporation and placed into plate readers at 30˚C for 48h without

agitation. $OD_{600}$ was read every 15 minutes at four points within the well, and mean $OD_{600}$ was used as a reading.

The growth rate for strain IGA130 with a plasmid-encoded DHFR was calculated from the slope at every time point before 40h, and taking the median of the five highest values for a more robust estimate. This was done for every replicate. Growth coefficient was measured for each replicate using the following equation (Eq 1), with the derivative growth rate of strain IGA130 with plasmid pAG416-GPD-Empty (Empty) without MTX as reference for maximum growth rate:

$$Growth\ coefficient = 1 - \frac{(Reference\ growth\ rate - Sample\ growth\ rate)}{Reference\ growth\ rate} \quad (1)$$

Inhibitory concentrations (ICs) were computed using the standard Hill equation, using Python's SciPy optimize.curve_fit method (SciPy version 1.7.3).

## Deep mutational scanning library preparation

A deep mutational scanning library of PjDHFR was constructed using the previously described megaprimer method [58,59]. Primers with NNK degenerate sequence at the codon of interest were ordered from Eurofins Genomics for all positions along PjDHFR, except for the start and stop codons. By using these primers with reverse primers Mega_1_R, Mega_2_R and Mega_3_R, for fragments 1, 2 and 3, respectively, megaprimers were generated by PCR using KAPA Hi-Fi polymerase and pAG416-GPD-PjDHFR as template on a per-position basis. This PCR product was diluted 1/2500 with two serial 50-fold dilutions, and 2 μL of these dilutions were used as megaprimers for the amplification of pAG416-GDP-PjDHFR using KAPA Hi-Fi polymerase. PCR products were digested using 0.2 μL of DpnI restriction enzyme (New England Biolabs, catalog #R0176L) (37˚C for 2 h) to remove parental DNA. This product was then transformed in *E. coli* MC1061, and colonies were counted. Biomass from petri dishes with >400 colonies was harvested (>10x coverage for each codon), and plasmid libraries extracted per position using the FroggaBio Presto Mini Plasmid Kit (catalog #PDH300.) Each plasmid positional library was quantified using a ThermoScientific Nanodrop 2000 spectro-photometer and library content was assessed for diversity using the RC-PCR method from [60] with primers Lib_seq_F1_F/R, Lib_seq_F2_F/R and Lib_seq_F3_F/R for the different fragments. Positions where diversity was too low were redone using the same protocol. All oligos used for library generation are available in S3 Table.

Once diversity was confirmed at every position, plasmid libraries were transformed using the same method as above into IGA130 and plated on SC-URA+MSG media. Biomass from Petri dishes with >1000 colonies were harvested, and glycerol stock cultures were prepared for each position. Once every position was transformed in yeast and appropriately frozen as glycerol stocks, 100 μL of each glycerol stock was added to individual 5 mL culture tubes of Sc-URA+MSG liquid media and grown overnight. $OD_{600}$ was measured for each position, and an equal number of cells from each position was mixed as a master pool for each fragment. A glycerol stock was prepared for each fragment.

## Pooled competition assay

Starting from these master stocks, 100 μL were added to 5 mL of SC-URA+MSG media and grown overnight in triplicate. These cultures were considered as $T_0$ samples. $OD_{600}$ was measured, and 1.5 $OD_{600}$ units were spun down at 224 rcf for 3 minutes and resuspended in 1 mL of ddH$_2$0. 14 mL of 1.07x MTX media was inoculated with 1 mL of 1.5 $OD_{600}$ culture, to yield 1.0x media at 0.1 $OD_{600}$, in triplicate. Depending on the condition, these cultures had either

0 μM (DMSO), 8.8 μM (IC 75) or 44 μM (IC90) of MTX. DMSO was used as the drugless control, as it is the solvent for MTX. Once cultures reached either 1.2 $OD_{600}$ or were grown for 72 h, cultures were diluted back to 0.1 $OD_{600}$ in a fresh batch of the same media using the same method. Once grown, these culturse were considered as $T_{final}$, and cells were harvested once cultures reached either 1.2 $OD_{600}$ or 72 h of growth in this second passage. At every passage and for every culture, triplicate glycerol stocks were made, and 5.0 $OD_{600}$ units were harvested and spun down at 224 rcf for 3 minutes. Supernatant was removed and pellets were frozen for downstream DNA extraction. DNA extractions were conducted using Zymoprep Yeast Plasmid Miniprep II (Zymo Research, catalog #D2004). The coding sequence of PjDHFR was split into three fragments of roughly equal length (F1, positions 2 to 73, F2, 74 to 145, and F3, 146 to 206) to allow Illumina MiSeq 300 bp paired-end sequencing. Extracted libraries were amplified by fragments using Lib_seq_F/R primers using a cycle described in S5 Table, purified using magnetic beads (AMPureXP from Beckman Coulter Life Sciences, catalog #A63882) and sent for sequencing for a total of 15M reads.

## Next-generation sequencing read processing

Reads were analyzed using custom Python and Bash scripts. Briefly, demultiplexed R1 and R2 reads were trimmed to remove primer sequences and merged using PANDAseq [61]. Merged reads were aggregated and counted using VSEARCH [62]. Using custom Python scripts, expected codon mutations were matched with read counts, and the frequency of each mutant was assessed in each individual culture. To minimize sampling bias, we ensured that every retained mutant had high initial coverage in $T_0$ to minimize bias in downstream analysis and discarded all mutants that had an initial coverage of less than 15 high quality reads. Synonymous codons to wild-type PjDHFR were used as internal control and reference for normalization between fragments. The selection coefficient was calculated according to Eq 2, where the normalized log2-fold change of the mutants between $T_0$ and $T_{final}$ was divided by the median normalized log2-fold change of the synonymous mutants and normalized by the number of mitotic generations measured for each replicate (g). The final selection coefficient value for a mutant was computed using the median selection coefficient between all synonymous mutants within each individual culture.

$$Selection\ coefficient = \frac{Log2\left(\frac{Freq\ mutant\ T_{final}}{Median\ freq\ silent\ T_{final}}\right) - Log2\left(\frac{Freq\ mutant\ T_0}{Median\ freq\ silent\ T_0}\right)}{g} \tag{2}$$

This normalization allowed us to generate a standardized score where 0 is the median score of synonymous mutants, which gives an accurate estimate of the wild-type fitness in the tested conditions and of the relative performance of all mutants in comparison to wild-type PjDHFR in every condition. The resulting distribution of selection coefficients was used in downstream analyses.

## Statistical analysis of selection coefficients and false-discovery rate

To assess resistance thresholds using statistical means, we used the sklearn mixture package (version 1.0.2) [63] to estimate the fraction of neutral mutations, resistant and very resistant mutations by fitting the distribution of selection coefficients under MTX selection to a mixture model of underlying Gaussian distributions. Using custom scripts, we estimated how many parameters recapitulated the underlying function the best by minimizing Akaike information criterion (AIC) and Bayesian information criterion (BIC) values for the gaussian mixture models using an expectation-maximization model. AIC and BIC are metrics to minimize to

favor the simplest model to maximize fitting. By using the best fit model, we separated our distributions of selection coefficients into their underlying Gaussian mixture models, and set our thresholds based on parameters from the two rightmost distributions (most resistant mutants and second-to most resistant mutants). To minimize overlap effects between underlying Gaussian distributions, we chose the median values of the second-to-highest underlying distribution as the resistant threshold, and the intersection second-to-highest and highest as the very resistant threshold in each condition where the drug was present (S1 Fig).

To correct for experimental noise and false discovery rate, we conducted a one sided Welch's *t*-test (scipy.stats version 1.7.3) [64]. We generated a matrix for each mutation at each position, which regrouped values for all replicates and synonymous mutations for a specific amino acid substitution and compared it to the distribution of silent mutations to assess if the selection coefficient of mutants with a substitution was significantly greater (i.e. more resistant) than the distribution of silent mutations. We then used the Benjamini-Hochberg method to control for the false discovery rate of significantly resistant mutants at 5% (statsmodels version 0.13.5) [65]. This allowed us to identify a subset of mutants that have selection coefficients significantly greater than the wild-type, which we classify as being resistant, and corrected for false discovery rate. All custom code is available at https://github.com/Landrylab/Rouleau_et_al_2023.git.

## Construction of validation mutants

To validate results from the pooled competition assay, we selected different mutants based on their scores, patterns of resistance, and reproducibility across replicates and conditions, and reconstructed them individually (S6 Table). To generate these mutants, identified by their specific reconstructed mutation in S2 Table, we used fusion PCR followed by Gibson assembly. First, by using oligonucleotides containing the mutation of interest that are a reverse complement to each other, we generated the two fusion PCR fragments using Valid_master_F and any R_valid oligo, and Valid_master_R and any F_valid oligo. pAG416-GPD-PjDHFR was used as the template vector. PCR products were mixed and diluted 1/2500, and used as a template with Valid_master_F and Valid_master_R oligos for the generation of the final product. The fusion PCR product was then used in Gibson cloning, using the same method and backbone as described above, and clones were validated using Sanger sequencing. Constructions were transformed into the relevant strain (IGA130 or FDR0001) and plated on SC-URA+MSG +/- 100 nM β-estradiol depending on the strain. Individual growth curves and growth rates were measured as described above. Subsequent spot-dilution assays were conducted as described above, but with 44 μM of MTX instead of 440 μM.

## Western blot and flow cytometry analysis

For western blot analysis, different variants of PjDHFR and ScDHFR were tagged using either 3xFLAG tag in C-terminal (3') or 1xFLAG tag in N-terminal (5'). For N-terminal constructions, pAG416-GPD-PjDHFR (wild-type, validation mutants pAG416-GPD-PjDHFR-M33P and pAG416-GPD-PjDHFR-F199P) or pAG416-GPD-ScDHFR were digested using ClaI enzyme in rCutSmart buffer (New England Biolabs, catalog #R0197) according to manufacturer's instructions, and purified using magnetic beads as described above. 3xFLAG cassettes were amplified using oligos C3xFLAG_F and _R from p3-3xFLAG-KanMX. The PCR product was purified using magnetic beads. Gibson assembly was conducted as described above and confirmed by Sanger sequencing using pAG_insert_F. For C-terminal tagging, 1xFLAG cassette was amplified from p3-3xFLAG-KanMX using N1xFLAG_F1_F and N1xFLAG_F1_R_-nomet or _met, to remove or keep the PjDHFR start codon, respectively. Then, PjDHFR was

amplified from pAG416-GPD-PjDHFR using N1xFLAG_F2_F_nomet or _met and N1xFLAG_F2_R, to remove or keep the PjDHFR start codon, respectively. These fragments were assembled using fusion PCR, as described above, and assembled into a digested (HindIII, rCutSmart, New England Biolabs, catalog #R0104) pAG416-GPD-ccdB using in-house Gibson assembly. All constructions were confirmed by Sanger sequencing using pAG_insert_F. All constructions were transformed in IGA130 and transformants were grown overnight in SC-URA+MSG+100 nM β-estradiol in triplicate. A schematic for all constructions is available in S2A Fig. The mEGFP tagged PjDHFR plasmids were constructed using the same procedure.

Yeast proteins were extracted by resuspending 12.5 $OD_{600}$ units in 250 μL of lysis buffer (Complete Mini Protease inhibitor cocktail, Millipore-Sigma 11836153001). Glass beads were added to the cell suspension and the tubes were vortexed for 5 min on a Turbomix. Following the vortexing step, 25 μL of SDS 10% were added to each tube before boiling them for 10 min. Once the tubes were boiled, they were centrifuged for 5 min at 16,000 rcf to clear the supernatant. Samples for migration were prepared by mixing 17.5 μL of cell extract with 2.5 μL of DTT 1 M and 5 μL of loading buffer 5X (250 mM Tris-Cl pH 6.8, 10% SDS, 0.5% Bromophenol Blue, 50% glycerol). Samples were migrated on a 15% SDS-PAGE acrylamide gel for 55 min, then transferred on a nitrocellulose membrane using a semi-dry system (TE 77 PWR, Amersham Bioscience) for 1h at 0.8 $mA/cm^2$. Following the transfer, membranes were soaked in a 1 mg/mL Ponceau red solution for 5 min, destained quickly with water and imaged. To remove remaining Ponceau staining, the membranes were shaken in PBS (10 mM sodium phosphate dibasic, 135 mM NaCl, 2 mM KCl, 1.5 mM monopotassium phosphate) solution for around 45 min. Membranes were then blocked overnight with agitation in blocking solution (Intercept (PBS) Blocking Buffer, Mandel Scientific, catalog #927–70003) at room temperature. The following morning, membranes were incubated with a primary antibody 0.15 μg/mL (Anti-Flag M2, F3165-1MG, Millipore-Sigma) in the blocking solution + 0.02% Tween 20 for 30 min. Membranes were washed 3 times for 5 min with PBS-T (PBS with 0.1% Tween 20). The secondary antibody was then added at 0.075 μg/mL (Anti-mouse 800, catalog #LIC-926-32210, Mandel Scientific) in the blocking solution + 0.02% Tween 20 for 30 min. Membranes were washed again 3 times for 5 min with PBS+T, followed by imaging with an Odyssey Fc instrument (Licor, Mandel Scientific) in channels 700 (30s) and 800 (120s).

### Flow cytometry assay

Mutants of PjDHFR-mEGFP were transformed in to strain IGA130 following the protocol described above. Flow cytometry measurements were conducted in triplicate for each mutant and control strain, with 5000 events per replicate per mutant. This was done using Guava's easyCyte BG HT with a 488nm excitation laser and a *525/530 nm filter*. Fluorescence intensity was corrected for front scatter, a measure of cell size, to normalize fluorescence intensity per cell, and transformed into log10 scale for visualization.

### Structural analysis, alignments, and allostery analysis

The structure of PjDHFR was recovered from AlphaFoldDB (UniProt: A0EPZ9). The crystal structures of DHFR ortholog from *P. carinii*, in complex with MTX/NADPH (PDB: 3CD2) and DHF/NADPH (PDB: 4CD2), were recovered from UniProt. Using ChimeraX's (version 1.6) [66] matchmaker algorithm, the structures were aligned with each other, and ligands from crystal structures were extracted and added to PjDHFR AlphaFold structure. The resulting complexed structures were then relaxed using the FoldX (version 5.0) optimize command to remove van der Waals clashes. Using the MutateX protocol on the PjDHFR complex with MTX/NADPH and DHF/NADPH, changes in protein free energy were computed [86, 87].

For visual representation of selection coefficient on protein structure, chains were colored using the ChimeraX color *byattribute* command, and attributes were manually defined within custom.defattr files. Data for coloring *E. coli* DHFR according to allostery was recovered from S1 Table from [67], with darker colors corresponding to more stringent confidence intervals for allostery. Alignments between *E. coli* DHFR and PjDHFR were done using ChimeraX's matchmaker.

To measure distance between protein residues and ligands, a custom script was used, based on tools from BIO.PDB (version 1.78) [68]. Briefly, by using a PDB file coordinate parser, distance was estimated between α carbons from all residues in PjDHFR and all atoms from the ligand of interest, except hydrogen atoms. Then, the minimal distance between the residue α carbon and atoms in the ligand of interest was considered for establishing contact. Any distance shorter than 8 Å was considered a contact.

## Modeling of binding affinity changes

To model changes in binding affinity to ligands caused by mutations, we used the Rosetta FlexddG method [69]. Briefly, binding molecules were parameterized using Rosetta's molfile_-to_params.py script. *In silico* saturation mutagenesis was conducted using a modified version of the provided script from the FlexddG GitHub tutorial, and the generated params files as extra param files [69]. 100 structures with 35000 backrub steps were used as parameters, based on recommendations from [70], and all individual resulting structures were analyzed using a modified version of the provided script from the FlexddG GitHub tutorial. This resulted in a distribution of ddG values for 100 individual replicates. To minimize extreme effects caused by potential clashes or other artifacts, the median values from these distributions were used as the final ddG for downstream analyses.

## Results

### PjDHFR can functionally complement the deletion of native DHFR, *DFR1*, in *S. cerevisiae* and is sensitive to methotrexate

In the budding yeast *S. cerevisiae*, the DHFR is encoded by the gene *DFR1* (ScDHFR). *DFR1* is an essential gene [71], which means that complementation assays require the use of an engineered strain. Some strains with *DFR1* deletion or downregulation have been constructed, but they require multiple mutations, such as inactivation of *TUP1*, and specific growth media to be handled [12,72,73]. Here, we performed the full deletion of *DFR1* in *S. cerevisiae* for functional complementation with PjDHFR. We engineered a strain in which the function of *DFR1* is complemented by introducing another DHFR-coding gene (*dfrB1*) from bacteria, that is non-homologous to *DFR1* and that is insensitive to antifolates [74]. Upon induction of a β-estradiol inducible promoter, expression of this *dfrB1* masks the deletion of *DFR1*, and allows for growth in the presence of MTX (Fig 1A and 1B) [75]. This strain, FDR0001 (*dfr1Δ*), is therefore non-viable in the absence of β-estradiol, but viable and MTX-resistant in the presence of β-estradiol.

By transforming FDR0001 with a plasmid harboring PjDHFR under the regulation of the constitutive GPD promoter and by growing it in the absence of β-estradiol, we found that PjDHFR functionally complements the deletion of *DFR1* in *S. cerevisiae* and rescues growth at a near wild-type level (Fig 1B). In the absence of β-estradiol, the strain without DHFR on the plasmid (Empty, negative control) was unable to grow, confirming that the phenotype observed depends on the presence of a complementing DHFR on the plasmid. Using media supplemented with MTX at a concentration known to fully inhibit *DFR1* at its endogenous

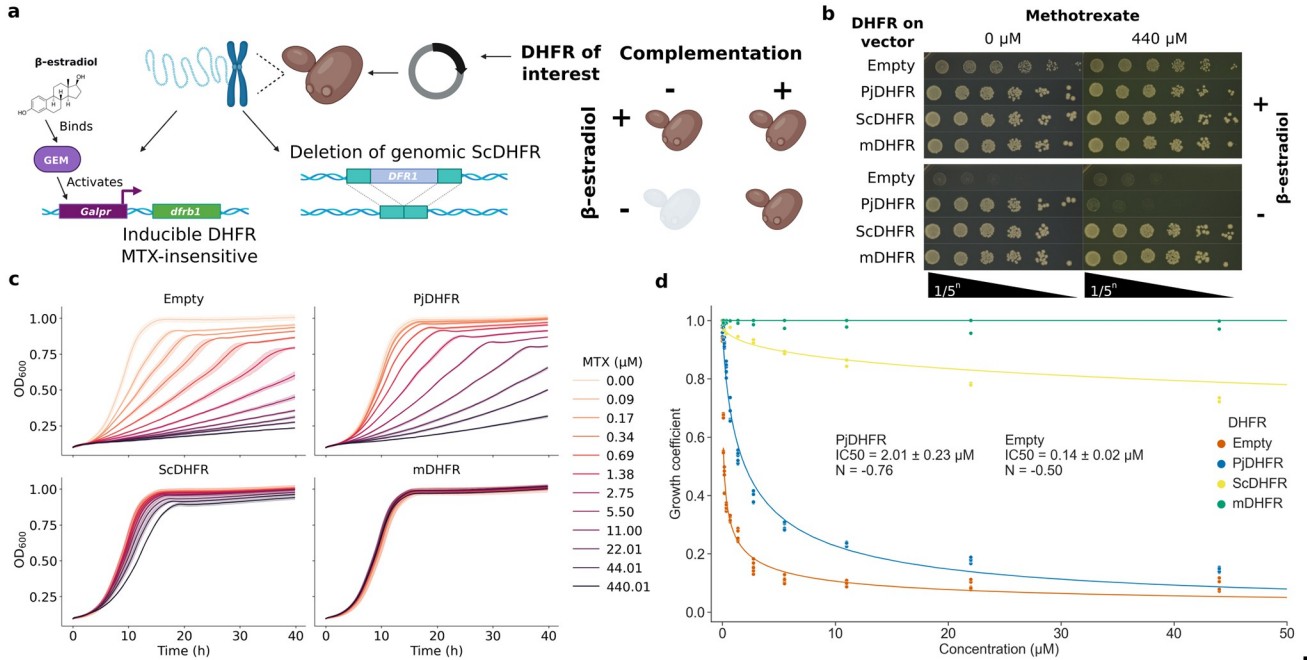

**Fig 1. Experimental setup for complementation assay using PjDHFR and dose response curves. a)** Experimental setup and strain construction. Yeast strain FDR0001 harbors the *dfrB1* gene under the regulation of a β-estradiol inducible promoter (*Galpr*) [75], and a *dfr1* deletion. *dfrb1* is insensitive to MTX, therefore all strains can grow on MTX-containing media in the presence of β-estradiol. This panel was made with BioRender. **b)** Spot-dilution assays to test for DHFR complementation and for MTX sensitivity. PjDHFR, ScDHFR and mDHFR are from *P. jirovecii* (GenBank: ABB84736.1, codon optimized for *S. cerevisiae*), *S. cerevisiae* BY4741 *DFR1*, and *M. musculus* mDHFR (synthetic, L22F and F31S, from [77]), respectively. ScDHFR was used as a complementation control, and mDHFR as a MTX-resistant control. When β-estradiol is added to the media, *dfrB1* is expressed, which is necessary for cell growth in a *dfr1Δ S. cerevisiae* background without DHFR on the vector. **c)** Growth curves of yeast strain IGA130 (wild-type for *DFR1*) each expressing a different DHFR. Shaded areas represent confidence intervals across biological triplicates. Empty vector shows sensitivity to low concentrations of MTX. Strains with vector-expressed DHFRs show increased resistance, with PjDHFR being the most sensitive, followed by ScDHFR, and mDHFR(L22F/F31S) being fully resistant. **d)** Growth coefficient (measure from growth rate relative to Empty without selection) of strain IGA130 with different DHFRs. $R^2$ for Hill equation fit: Empty = 0.96, PjDHFR = 0.98, ScDHFR = 0.91 and mDHFR = 0.

expression level (440 μM) [76], we used this same system to assess PjDHFR sensitivity to this antifolate. The MTX-resistant mouse DHFR (mDHFR) showed complementation and resistance to MTX, as expected [77]. Complementation of the inhibition of *DFR1* by this mDHFR in *S. cerevisiae* and its associated resistance to MTX have been reported in several studies [76,77]. Interestingly, when using a plasmid harboring ScDHFR under the regulation of the GPD promoter as a positive control for complementation, we observed resistance to 440 μM of MTX on solid media. Since *DFR1*, at its native expression level and without vector-expressed DHFR (Empty) is sensitive to MTX, this experiment shows that high expression of this protein is sufficient to achieve resistance to MTX at these concentrations. Resistance caused by overexpression has previously been reported in *E. coli* for the antifolate TMP [78]. Despite being under the regulation of the same promoter, the PjDHFR-harboring strain was sensitive to MTX. Western blot results (S2B Fig), alongside complementation assays, show that PjDHFR is expressed and functional in *S. cerevisiae*, and that PjDHFR is more sensitive to this antifolate than wild-type *DFR1*, at least in this context. This highlights the potential for PjDHFR to evolve toward resistance to MTX and the usefulness of our system to identify resistance mutations.

To quantitatively assess the extent of inhibition and calculate inhibitory concentrations (IC) of PjDHFR in *S. cerevisiae*, we measured growth curves in the presence of decreasing

concentrations of MTX. By comparing growth rates measured in the presence of different concentrations of the drugs to the growth from the Empty vector in the absence of the drug as growth coefficient, we estimated the sensitivity of strain IGA130, which still has wild-type *DFR1* at the genomic locus. Here, we switched from strain FDR0001 (*dfr1Δ*) to the parent strain IGA130 (wild-type for *DFR1*) for downstream screening because FDR0001 had insufficient transformation efficiency for the construction of DMS libraries. Indeed, despite media supplementation with β-estradiol, strain FDR0001 shows growth defects when compared to strain IGA130, which reduces transformation efficiency. We believe that this growth defect creates a selection for fully functional PjDHFRs mutants, and that non-functional or less-functional mutants could be lost at this step. As we focus on mutations that confer resistance to MTX, we wanted to limit pre-screening selection on PjDHFR mutants.

In strain IGA130, mDHFR and Empty vector were used as positive and negative controls for MTX resistance, respectively (Fig 1C). None of the tested concentrations had a detectable effect on the growth rate of strains harboring MTX-resistant mDHFR. Empty plasmid showed the highest sensitivity, as only the natively expressed *DFR1* was present in this strain. Using a Hill-equation curve fit, we calculated ICs for these different DHFRs (Fig 1D). IC50, when expressing no exogenous DHFR, was over 10 times lower than when expressing PjDHFR (0.14 μM and 2.01 μM, respectively), highlighting the sensitivity of the genomic *DFR1* and the relatively limited impact of genomic *DFR1* on resistance to MTX. The conditions that were selected for downstream competition assays correspond to roughly 75% and 90% inhibition for PjDHFR (IC75 = 8.8 μM MTX and IC90 = 44 μM MTX).

To ensure that the use of strain IGA130 (wild-type for *DFR1*) did not introduce a bias in selection strength, we conducted identical dose response curves in strain FDR0001 (*dfr1Δ*) and estimated its IC50 as well (S3A and S3B Fig). IC50 measured for strain FDR0001 is of 1.95 μM, validating results from strain IGA130 on the minimal effect of genomic *DFR1* on MTX resistance. We believe that by using this system, we were able to identify a maximum number of mutations in PjDHFR that confer resistance to MTX, without introducing downstream bias for PjDHFR activity.

## Pooled competition assays of PjDHFR mutant library identify mutations conferring resistance to methotrexate

Deep mutational scanning (DMS) was conducted on PjDHFR using the megaprimer method [59]. We generated all possible single amino acid changes of PjDHFR on the plasmid described above. To assess the fitness effect of mutations, we measured the change in frequency for every mutant between the initial culture ($T_0$) and the second passage in the presence of MTX ($T_{final}$). The pooled competition assay in DMSO was included to identify any mutations that could have dominant negative effects on growth and that could thus confound resistant or sensitive phenotypes (Fig 2A). IC75 and IC90 (Fig 2B and 2C) selection coefficients were used to identify mutations that cause resistance to MTX. The final count of mutants covered 97.2% of all possible amino acid replacements (4166/4284). We calculated a selection coefficient for each mutant in each condition, which quantifies the effect on growth of an amino acid substitution relative to the wild-type PjDHFR, where a positive score represents a growth advantage, and a negative score a growth disadvantage. Replicates within conditions were well correlated (S4A, S4B and S4C Fig).

By using results from the competition assay with MTX, we identified mutations in PjDHFR that confer resistance to this antifolate. To classify mutants as being resistant or not, we first used a Gaussian mixture model to separate the distribution of fitness effects into the number of Gaussian components that best recapitulate the overlying distribution of observed fitness

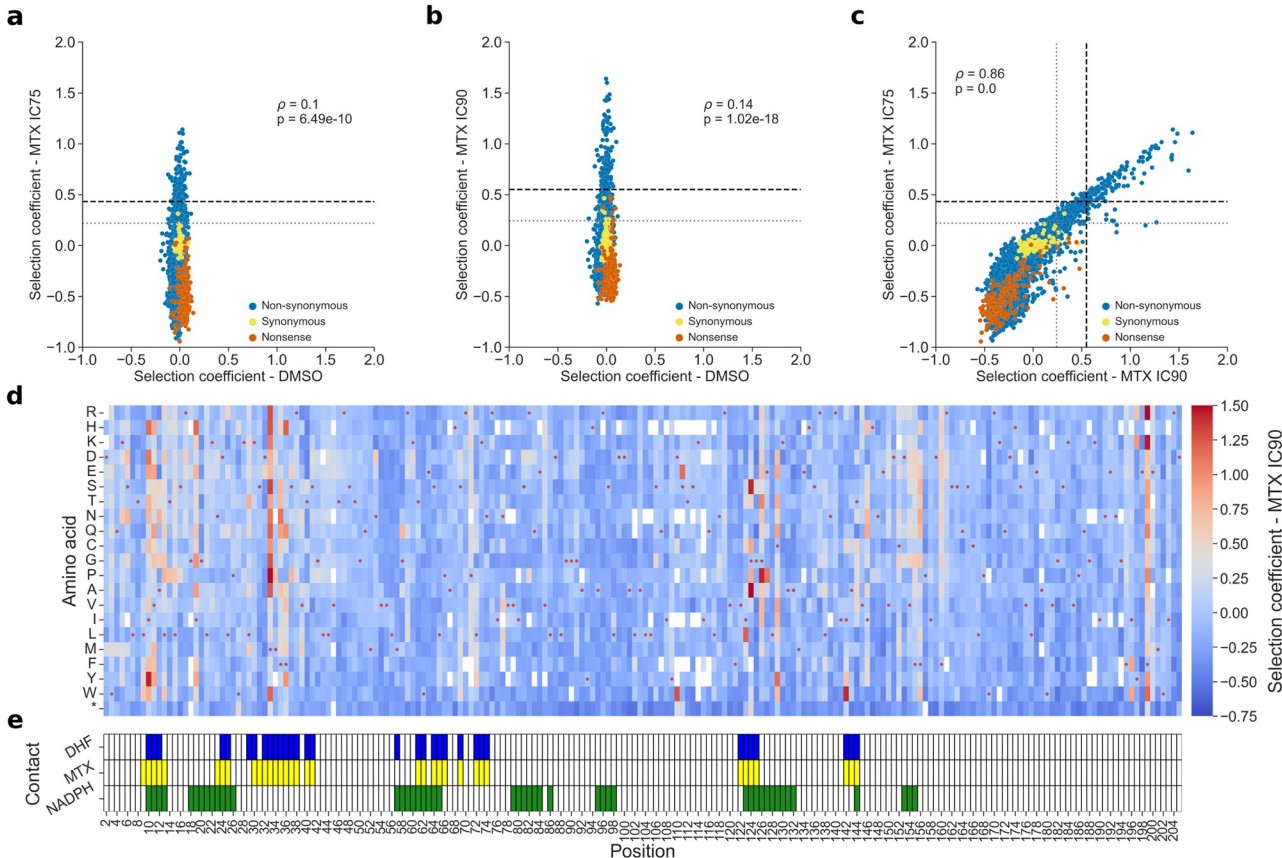

**Fig 2. Effects of amino acid substitutions on the resistance of PjDHFR to methotrexate. a)** Comparison of selection coefficients between control condition (DMSO) and at IC75 (Spearman's $\rho$ = 0.1, p = 6.49e-10), **b)** DMSO and IC90 (Spearman's $\rho$ = 0.14, p = 1.02e-18), and **c)** IC75 and IC90 (Spearman's $\rho$ = 0.86, p < 1e-100). The different types of variants found in the library are marked. Nonsense mutants have premature stop codons. Synonymous are mutants that have different codon sequences but that have the same amino acid sequences as the wild-type sequence. Non-synonymous are amino acid changes relative to wild-type PjDHFR. Resistant (grey/dotted line) and very resistant (black/dashed line) thresholds are shown for both MTX conditions (See *Statistical analysis of selection coefficients* in methods for threshold determination, S1 Fig). **d)** Selection coefficient of a given amino acid (y-axis) at a given position (x-axis) at a MTX concentration corresponding to 90% of growth inhibition (IC90). Red points show the wild-type sequence of PjDHFR. **e)** Positions of contacting residues along PjDHFR modeled on structural alignments between PjDHFR and orthologous *P. carinii* DHFR (PDB: 3CD2 (MTX and NADPH) and 4CD2 (DHF)). Contact was established as amino acids with an α carbon located less than 8 Å from MTX, DHF or NADPH. Detailed heatmaps for DMSO and IC75 are available in S5 Fig.

effects at IC75 (five components) and IC90 (four components) (S1 Fig). By using the estimated parameters from the underlying functions, we set the thresholds for resistance using quantiles of the distributions centered on a selection coefficient of 0. For IC75, the thresholds for a resistant mutant and a very resistant mutant were set at selection coefficients of over 0.220 (404/4284 mutants, 9.43%) and 0.432 (129/4284, 3.01%), respectively. For IC90, 0.243 (437/4284, 10.20%) and 0.550 (134/4284, 3.13%), were set for resistant and very resistant, respectively. Because of the difference in the number of modelled Gaussian function components in the two conditions, potentially caused by the difference in strength of the selection pressures, slight differences in the number of mutants that pass the different thresholds are expected.

Overall, 31 positions, or about 15% of the positions along the protein, had at least one mutant that passed the very resistant threshold in both concentrations of MTX. These cutoffs do not consider the various numbers of replicates per amino acid in these experiments, so mutations above these thresholds do not have the same levels of confidence. We therefore also

conducted a one-sided Welch's t-test and corrected for false discovery rates using the Benjamini–Hochberg procedure to control for the false discovery rate of significantly resistant mutants at 5%. We tested whether selection coefficients for amino acid variants were significantly greater than that of synonymous mutants of the wild-type PjDHFR. For IC75, 836 mutants with selection coefficients >0 were considered as significant, and 694 for IC90, with an intersection between the two concentrations of 603 mutations (S1C and S1D Fig, green line, and S6 Fig). Finally, analyzing intersecting datasets from thresholds and statistical significance, we identified 355 mutants as being resistant with a high level of confidence, and 116 mutants that we consider very resistant with a high level of confidence.

## Individual growth assays validate results from the pooled competition assay

To validate the results of the pooled competition assays, we manually reconstructed 44 mutations in PjDHFR (Fig 3A, 3B and 3C). These mutations were chosen because they had various selection coefficients along the gradient of values observed. We also included positions that showed discrepancies between the IC75 and IC90 conditions, such as G58Q, D110W, S111E, I196W and I196F (Fig 3A, highlighted green). These mutants exhibited high resistance at IC90 but behaved differently at IC75, not passing the threshold for very resistant. Because the results strongly correlate between these two conditions, we reasoned that these could be experimental outliers, as we had observed increased noise between replicates at IC90 compared to IC75 (S4B and S4C Fig). These outlier mutants are enriched for amino acids encoded by a single codon in our experiment, which could partly explain their higher noise. Validation mutants were tested using growth curves and spot-dilution assays at IC90 in the same conditions as the competition assay, as well as in DMSO. Results of the growth assays confirmed that these outliers were indeed sensitive to MTX at IC90 (Fig 3B). Growth rate for all other mutants individually tested correlated strongly with their selection coefficient from the competition assay based on Spearman's rank correlation (Fig 3B and 3D). All mutants grew similarly in DMSO (Fig 3C and 3D).

To further validate our results and assess the presence of tradeoff between function and resistance, we reintroduced some of the reconstructed validation mutants into strain FDR0001 to assess the mutant protein's potential for functional complementation. In the absence of β-estradiol, all mutants identified as resistant could complement the deletion of *DFR1* in this strain, as expected. Interestingly, most mutants which were identified as being sensitive lost their ability to complement the deletion, implying that these mutations strongly negatively impacted protein function (S8A Fig). In the presence of β-estradiol, all mutants grew similarly in the presence or absence of MTX (S8B Fig). Growth experiments in strain FDR0001 exhibited higher levels of noise than similar experiments in IGA130 (Fig 3A vs S8C and S8D Fig error bars).

We then used these results to correlate growth rate in IGA130 and in FDR0001 to validate how the results from the high-throughput assay could be transposed to strain FDR0001 (*dfr1Δ*). In DMSO, all mutants in strain IGA130 grew similarity, but non-functional PjDHFR mutants in strain FDR0001 were not able to grow. In the presence of MTX, the phenotypes of resistant mutants are very well correlated between the two strains, highlighting the robustness of this assay (S8C and S8D Fig).

## Premature stop codons reduce gene dosage toxicity

In the absence of MTX, the distribution of selection coefficients showed little variance around 0. As the endogenous *DFR1* was present in strain IGA130, the effect of PjDHFR mutants on

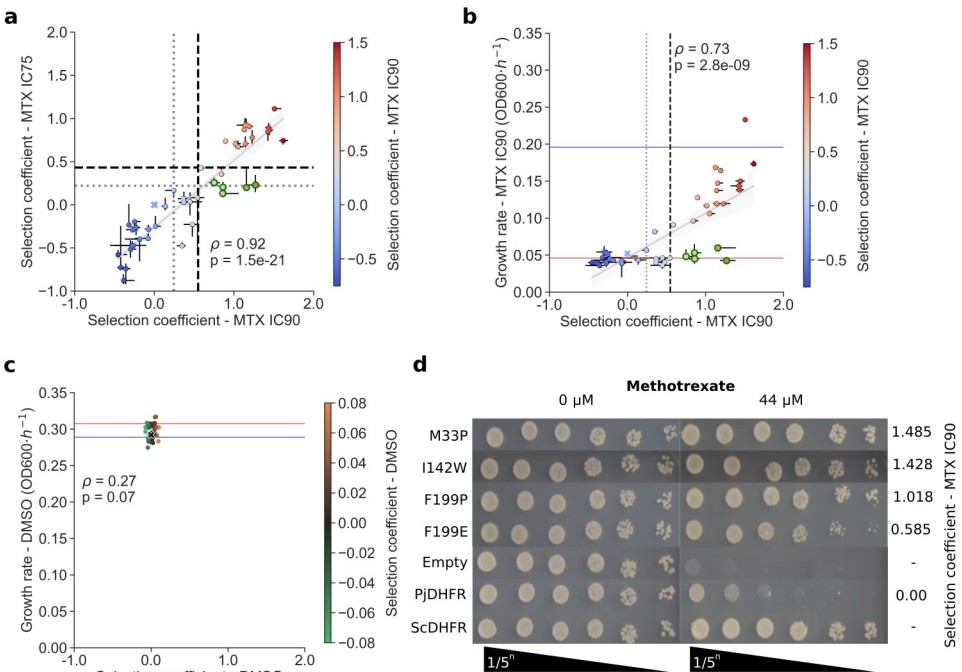

**Fig 3. Validation of pooled competition assay with individually reconstructed mutants. a)** Selection coefficient at IC75 and IC90 for selected mutants. Variants at positions showing large deviations between IC75 and IC90 are visible in the bottom right quadrant (gray lines, resistant threshold, black lines, very resistant thresholds, from gaussian mixture models, S1 Fig). Wild-type PjDHFR (blue X) is at 0.0 selection coefficient. Error bars in x represent mutant selection coefficients across synonymous codons and replicates at IC90 MTX and in y, at IC75 MTX. **b)** Growth rate ($OD_{600} \cdot h^{-1}$) compared to selection coefficient in IC90 for validation mutants. The blue line represents the derivative growth rate for the strain with ScDHFR on the plasmid and the red line represents the growth rate for the strain with the Empty plasmid (negative control). Error bars in x represent mutant selection coefficients across synonymous codons and replicates, and error bars in y represent growth rate measurements in triplicate. Suspected outliers are outlined in green. **c)** Growth rate compared to selection coefficient in DMSO for selected mutants. All mutants grew at similar rates in this condition. All selection coefficient axis limits are scaled to ease comparison between panels a), b) and c). Individual growth curves are shown in S7 Fig. Statistical test in a), b) and c) is Spearman's rank correlation. **d)** Spot-dilution assay for controls and some resistant mutants with their respective selection coefficients in IC90. ScDHFR and Empty are used as positive and negative controls, respectively.

growth without MTX was expected to be minimal. However, we observed an enrichment of positive selection coefficients for nonsense mutations (S5A Fig, row marked with *, S9 Fig) in this condition. Gene dosage toxicity has previously been reported for DHFRs at expression levels higher than wild-type, suggesting that this enrichment is the result of a fitness increase caused by the interruption of functional PjDHFR production [77,79]. Because early stop codons are more likely to prevent the formation of a functional protein than downstream ones, this effect is mainly observable at the beginning of the sequence and is lost by the end of the protein, with selection coefficients becoming slightly negative at around position 160 (S8 Fig). Other mutations, especially before position 50, with similar fitness effects are observable across the protein, which could hint at mutations that affect the function or folding of PjDHFR, preventing gene dosage toxicity. However, the effect of most such mutations (mean across all positive mutations in DMSO = 0.023, median = 0.016) is generally smaller than the mean of stop codons before position 160 (0.060, median = 0.059), with only 0.26% (11/4284) of mutations scoring significantly higher than this mean across the entire protein. When using FoldX to predict the effect of mutations on the stability of PjDHFR, we did not find a

significant correlation between the effect of mutations and the selection coefficient in DMSO (S10 Fig). We expected highly destabilizing mutations to be enriched in DMSO, as they would have potentially affected protein folding and thus catalytic activity. The absence of clear positive correlation implies that a higher selection coefficient in DMSO cannot be explained solely by changes in the protein's free energy. While the observed effects were small, the detection of the effect of stop codons highlighted the sensitivity of this type of competition assay. Indeed, when manually validating mutants from the competition assay using growth curves and spot-dilution assays, we did not detect this phenotype, although the strain with the Empty plasmid had slightly higher growth rate than any other wild-type DHFRs in this assay (in DMSO, Empty has a growth rate of 0.308 $OD_{600}{\cdot}h^{-1}$, and ScDHFR of 0.289 $OD_{600}{\cdot}h^{-1}$, Fig 3C red and blue horizontal lines, respectively). Most importantly, the experiments in the absence of MTX show that there are no major dominant negative effects of amino acid substitutions that would confound the measures of sensitivity to MTX.

## Resistance mutations most often appear at position within ligand binding pocket

Since there is no available crystal structure of PjDHFR, the predicted protein structure was recovered from AlphaFoldDB (UniProt: A0EPZ9). Crystal structures of the DHFR ortholog from *P. carinii* (61% identity, 75% positives), in complex with MTX/NADPH (PDB: 3CD2) and DHF/NADPH (PDB: 4CD2), were recovered from the Protein Data Bank (PDB) and aligned to generate the holoprotein structure of PjDHFR with MTX/NADPH and DHF/NADPH. Despite sequence identity of only 61% between the orthologs, the structures were highly conserved (root-mean-square deviation (RMSD) of 0.644 Å across the protein backbone heavy atoms, 2.094 Å across all heavy atom pairs), allowing for high-confidence structural alignments of ligands without the use of *ab initio* docking. We then mapped the selection coefficients on the structure of PjDHFR (Fig 4A and 4B).

Most resistance positions are located within the PjDHFR substrate binding pocket, or are in contact with NADPH, as we would expect for a competitive inhibitor such as MTX. These sites overlap with residues that bind MTX. Not all mutations at these positions lead to resistance. Indeed, resistance implies that the enzyme is no longer inhibited by MTX but maintains proper interactions with the substrate so that it can still perform its catalytic reaction that is required for growth. This leaves very few sites that can be modified. At some of these key positions, such as M33 (4.8 Å from MTX), resistance appears to also correlate with amino acid properties, where hydrophobic amino acids, such as the wild-type methionine, have generally lower selection coefficients in comparison to other polar residues, such as hydrophobic M33F (0.369 in IC90, p-value = 1.5e-8, from one-sided >0 Welch t-test), M33Y (0.243, p-value = 0.01) and M33V (0.441, p-value = 1.1e-5) compared to charged M33R (1.287, p-value = 8.8e-25), M33K (1.088, p-value = 4.7e-7) and short M33G (1.262, p-value = 1.3e-13). At other positions, fewer amino acids had positive selection coefficients. At I142 (7.1 Å from MTX), only two amino acid substitutions, I142W (1.428, p-value = 4.7e-4) and I142Y (0.668, p-value = 2.5e-3), caused resistance, implying that other position-dependent properties, such as side chain bulkiness, are at play. I142W, as well as a sensitive mutation at this same position, I142G (-0.167, p-value = 0.97), were manually reconstructed as part of the validations by growth curves and spot-dilution assays.

Another region where strong effects were observed was the glycine triplet at positions 124 to 126, which is conserved across species (3/3 in humans, *S. cerevisiae* and *E. coli*, 2/3 (positions 124 and 125) in mice). All three glycines from this triplet are in contact either with both MTX/DHF and NADPH (124 and 125) or with NADPH only (126), which could imply a role in

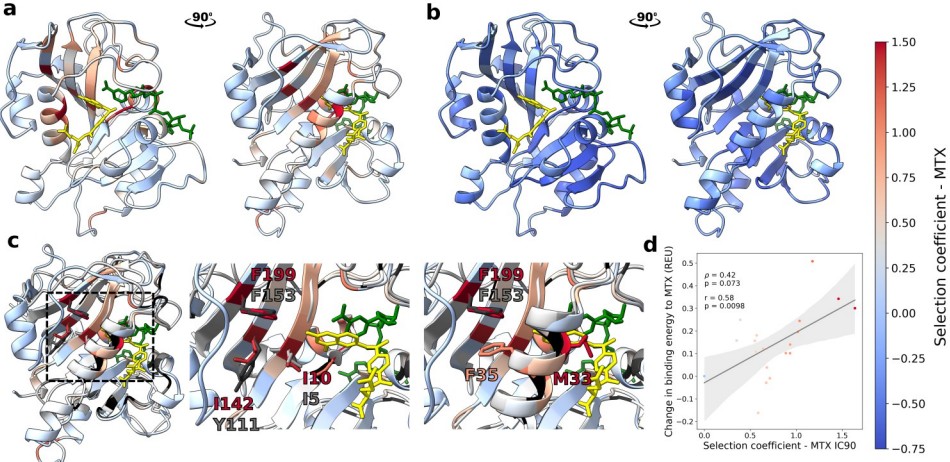

**Fig 4. Selection coefficients mapped onto PjDHFR structure in complex with methotrexate and NADPH. a)** Maximum selection coefficient from IC90 and **b)** minimum selection coefficient from IC75 observed for amino acid substitutions mapped on the PjDHFR structure by position. MTX (yellow) and NADPH (green) are visible. Nonsense mutations were not considered when identifying minimum selection coefficients. **c)** Structure superposition of PjDHFR (blue/red scale) and *E. coli* FolA (gray/black scale). FolA is colored according to predicted allostery from [67], with darker shades of gray representing increased confidence in predicted allosteric score. PjDHFR is colored by maximum selection coefficient in IC90. Orientations of superimposed structures are the same as panel A. Positions 199 and 142 are predicted to have allosteric effects on the substrate and cofactor binding pocket [67]. I10 is not predicted to have an allosteric effect. Side chains for these positions, and corresponding in *E. coli.*, are shown and labeled. **d)** Rosetta FlexddG predicts that mutations at position 199 would destabilize interactions between PjDHFR and MTX, and this effect correlates with the measured selection coefficient. Spearman's rank coefficient and Persons' correlation coefficient were used to measure correlation. Y axis units are REU (Rosetta Energy Units).

protein activity and/or ligand binding. Interestingly, at IC75 and IC90, mutants at position 124 with a positive selection coefficient generally have short chain amino acids (G124S/A/P 0.865/0.736/0.244 in IC75, p-values = 8.5e-13/4.8e-8/3.6e-7). Position 126 is less constrained, with 12/20 substitutions having a selection coefficient above the resistance threshold. However, at IC75, position 125 was one of the least permissive positions observed in the competition assay, with the lowest median score by position (-0.735), and no substitution with a selection coefficient higher than 0, implying that the glycine at this position is under strong purifying selection in this protein.

Mechanisms of DHFR resistance to MTX have previously been investigated in eukaryotes, mainly in baker's yeast, mice, and humans [77,80,81]. We examined how conserved the mechanisms were PjDHFR (S7 Table). Positions with mutations that are most often reported as conferring resistance to MTX, namely M35 in yeast, F31, F34 and Q35 in mice and in humans, are structurally equivalent to positions M33 (maximum selection coefficient at this position in IC90 = 1.485, p-value = 2.8e-11), F36 (maximum = 1.202, p-value = 2.1e-5) and S37 (maximum = 0.760, p-value = 3.1e-8) in PjDHFR, all being <5 Å from MTX. In PjDHFR, some mutations have been reported as conferring resistance to TMP in clinical isolates, with position F36 being reported as conferring the highest resistance to this drug [82]. This reveals that resistance mechanisms are highly conserved among DHFRs. However, while some mechanisms are conserved, and cross resistance is observed, other sites do not share these properties. One such position is position L65, which was reported as conferring high resistance to TMP in clinical isolates, and no mutations at this site were identified as resistant in this screen [38]. Other mutations that were reported or predicted to confer TMP resistance, such as mutations in the Q172-V181 loop (compiled in S7 Table), did not confer resistance to MTX in this study

[38]. As antifolates are competitive inhibitors, we expect most mutations that confer resistance to come with a tradeoff in protein activity. In mice and in humans, biochemical properties were measured for some mutants at position F31, structurally equivalent to position M33 in PjDHFR. Resistance was associated with an increased $K_m$ for DHF, which translates to a less active enzyme, up to four-fold for some mutants at this position, consistent with a tradeoff between resistance to MTX and activity [77,81,83].

## Potential mechanism of resistance through allostery

By mapping the maximum selection coefficient values at IC90 (Fig 4A) and minimum values at IC75 (Fig 4B) by position on the structure, we can visualize structural determinants of resistance. Most positions where at least one mutant is highly resistant are proximal to MTX/DHF and NADPH, or close to other positions where resistance is observed (Fig 4A). Inversely, almost every position (202/204) has at least one mutant that performed worse than the wild-type PjDHFR at IC75, except for positions M33 and F199. This is consistent with the expectation that most positions along a protein can have a deleterious effect if mutated, especially in the presence of strong selection like MTX [74].

While some mutations were consistent with the literature and structural properties of eukaryotic DHFRs, other resistant mutations were unexpected and may thus reveal new mechanisms of resistance. F199, a position located in a distal β-strand, and which has no direct contacts with drug, substrate, or NADPH, showed high levels of resistance to MTX for many different substitutions, and consistently so across both conditions. The maximum selection coefficients are 1.11 at IC75 and 1.64 at IC90, the second and first highest in their respective assays. Two mutants from this position, F199E (0.585 at IC90) and F199P (1.018 at IC90), were individually reconstructed by site-directed mutagenesis, and were validated for resistance to MTX (S7 Fig). While this position had not been reported in eukaryotic resistance studies, a structurally equivalent position (F153) had been predicted in *E. coli* as being part of the allosteric "sector" of FolA (PDB: 1RX2), which could hint at a yet unreported mechanism of resistance to MTX [67]. One mutation at this position, F153S, has also been identified in *E. coli* as slightly increasing resistance to TMP [78]. Interestingly, positions I10 (I5 in *E. coli*, I11 in baker's yeast, I7 in mouse and humans) and I142 (Y111 in *E. coli*, L139 in yeast, F134 in mouse and human) are two other positions where high levels of resistance were observed. Both positions are part of the same β-sheet and physically close to F199, forming a "line/groove" towards the α-helix containing M33, another position where several mutants had increased resistance to MTX, and which is in direct contact with MTX and DHF. Y111 in FolA, the structurally equivalent position to I142, has also been identified as being part of the allosteric "sector" of FolA [67]. All three positions, I10, I142 and F199, are relatively buried, and are hydrophobic amino acids, a property that is conserved across *E. coli*, baker's yeast, mouse, and humans. In the case of F199, amino acids with similar properties have lower selection coefficients in MTX, and polar/longer amino acids have higher selection coefficients, with F199R and F199K being respectively the first and fourth mutants with the highest selection coefficients at IC90. Position F199 is also in close proximity to the side chain of position F35, and could be forming π-stacking interactions with this other phenylalanine, which could then affect the α-helix where position M33 is located [84]. Position 199 also displays unique properties in DMSO, being the only position where a mutation, F199P (0.102, p-value = 0.22), is enriched, but the stop codon is not, implying that F199P is potentially more disruptive for protein function or folding than a nonsense mutation at this position. However, this is based on a single datapoint with three replicates and could therefore be due to experimental noise. Overall, this evidence points

towards a more important role of this position in DHFR function and resistance than previously estimated.

To investigate the specific effects of mutations at position 199, we conducted an *in silico* mutagenesis at this position using Rosetta's FlexddG method and predicted changes in binding energy between PjDHFR, MTX and DHF (Fig 4D) [69]. The changes in binding energy predicted by Rosetta FlexddG between PjDHFR and MTX show a slight positive correlation ($\rho$ = 0.42, Spearman's p-value = 0.07, r = 0.58, Pearson's p-value = 0.0098) with the selection coefficients measured at IC90 for each corresponding mutant, implying that mutations that destabilize interactions with MTX are more likely to be highly resistant (Fig 4D). Interestingly, FlexddG predicted that mutations at position 199 would generally be more destabilizing between PjDHFR and MTX than between PjDHFR and DHF, which would translate to a more resistant but still functional enzyme (S11 Fig). Most mutations that are more destabilizing correlate with measured selection coefficient. In *E. coli*, biochemical properties of mutant F153S were investigated. It was found to have no significant effect on the maximum rate of catalysis ($k_{cat}$) of FolA, but did increase $K_m$ to TMP, which is consistent with the decrease in binding affinity modeled by Rosetta [78]. This evidence points towards position 199 having distal effect on binding affinity, but not necessarily on catalytic rate, suggesting that mutants at this position could lead to resistance without strongly compromising enzyme function, making this position a good candidate for the evolution of resistance mutations.

## Alternative mechanisms of MTX resistance in PjDHFR

Alternative mechanisms of resistance were also investigated. Indeed, antifolate resistance caused by increased DHFR expression or protein abundance has been documented. To verify this, we constructed PjDHFR-mEGFP fusion proteins for wild-type PjDHFR and some validation mutants of interest. Fluorescence intensity was measured by flow cytometry as a proxy for protein abundance (S12A Fig). As all mutants are under the regulation of the same promoter, variation in abundance should depend on changes in protein stability caused by the mutations. We did not observe a correlation between resistance and protein abundance (S12B Fig). Mutants I10H and I10R (selection coefficients of 1.154 and 0.014, respectively) both had reduced abundance (0.71- and 0.82-fold change in relative fluorescence intensity per cell, respectively), where the most resistant mutant of the two is the least abundant. As for position 199 specifically, we had only reconstructed two validation mutants (F199P and F199E, selection coefficients of 1.018 and 0.585, respectively, both resistant since no mutations at position F199 leads to a negative selection coefficient). These two mutants showed slightly increased expression levels, but the measured increase in expression is not proportional with the measured selection coefficient (F199P and F199E, with increase of 1.05 and 1.02-fold-change in relative fluorescence intensity per cell, respectively). Based on these observations, we believe that MTX resistance cannot be solely attributed to increase in protein abundance, but it is possible that it is a contributing factor.

As the screen was conducted in strain IGA130 (wild-type for genomic *DFR1*), it was also possible that the expression of PjDHFR mutants from the plasmid caused fluctuations in the expression of genomic *DFR1*, affecting measured DMS scores. To test this, we constructed strain FDR0003, which is the same as strain IGA130, but where genomic *DFR1* is tagged with 3xFLAG-tag in C-terminal. We measured changes in genomic *DFR1* expression by anti-FLAG western blot in the presence of several resistant and sensitive mutants, and did not observe any changes in *DFR1* expression in the presence of PjDHFR mutants (S12C Fig).

## Discussion

### Functional complementation and DMS are powerful tools to investigate hard-to-study organisms

The study of *P. jirovecii* has always been difficult because of our current inability to cultivate it *in vitro* or in animal models. As resistance is increasing in this fungal pathogen, there is a pressing need to understand the mechanisms behind its emergence. By employing functional complementation in *S. cerevisiae* and a DMS library of PjDHFR, we investigated resistance to an antifolate using an exhaustive library of mutants. While this system does not allow the direct investigation of TMP resistance, we believe that in the light of the clinical and evolutionary link between MTX, TMP and *P. jirovecii* infection, this study provides relevant insights in the evolution of antifolate resistance, and on the biophysics of MTX resistance in DHFRs.

We demonstrate that under the regulation of a constitutive promoter, PjDHFR is sensitive to MTX in a quantitative manner, even in the presence of wild-type genomic *DFR1*. From this data, we targeted two selection pressures to conduct DMS followed by a competition assay to assess the effect of mutations compared to the wild-type PjDHFR. This allowed us to identify a broad range of fitness effects, both positive and negative, in the presence of MTX. We also observed mutations that affect protein function and gene dosage toxicity. Results from the competition assay are consistent with known resistance mutations to MTX in eukaryotes, implying that some resistance mechanisms are conserved across distant lineages. While individual validations in strain FDR0001 (*dfr1Δ*) allowed to identify resistance/function tradeoffs, the high-throughput screening in strain IGA130 (wild-type for *DFR1*) did not allow systematic investigation of this relationship. Further experiments will be necessary to understand the functional tradeoff associated with MTX resistance in PjDHFR.

By using the DMS library from PjDHFR to conduct competition assays in MTX at two different concentrations, we were able to measure different ranges of effect on resistance. Using higher selection pressure, such as IC90, allowed mutations that caused higher-resistance mutations to be more enriched in-regards-to wild-type-like mutations and showed a broader range of effect for highly resistant mutations. At a weaker concentration (IC75), mutations that performed worse than wild-type were more easily identifiable than at IC90, which is indicative of either mutants less resistant than the wild-type, or non-functional enzymes. Finally, by conducting the experiment in the absence of MTX selection, we were able to assess the absence of dominant negative effect, as well as to identify the presence of gene dosage toxicity through the effects of nonsense mutations. Combining the results of both datasets in the presence of MTX also helped identify mutants with inconsistent effects and guided the subsequent validation work. Validations in strain FDR0001 allowed to show a correlation between measured growth rates and scores with validations and high-throughput screening in strain IGA130 and highlighted the robustness of the system.

### Resistance to methotrexate occurs through different mechanisms

Most mutations that confer resistance to MTX were within the substrate or cofactor binding pocket, and most often, several different amino acid substitutions allowed for resistance at a given position. In the case of M33, shorter chain, hydrophobic amino acids (wild-type-like) were the substitutions that caused the lowest levels of resistance, implying that changes in local amino acid properties are key for resistance, instead of the specific amino acid present at this position. While M33 is in direct contact with MTX, F199 is not, but both displayed this pattern of "property-based" resistance. Other positions where resistance was observed, such as I10, displayed no such clear pattern. While protein abundance measurements showed no clear

correlation with resistance in this study, increase in protein stability has been well documented as an antifolate resistance mechanism, and at least two of the tested resistant mutants showed increase in protein abundance [77]. This implies that several mutational pathways are available for evolution towards resistance to MTX, and possibly other antifolates. This raises major concern about the future of resistance to antifolates in *P. jirovecii*. However, as MTX is not used to treat *P. jirovecii* infections in clinical settings, and because of the conflicting reports of the effects of MTX on the evolution TMP resistance and on *P. jirovecii* infection, it is important to be critical about the nature of these results. We believe, however, that this is a step in the right direction, and that this data will help future studies focusing on more clinically relevant antifolates.

Structural analysis and binding affinity modeling of the observed fitness effects allowed us to identify putatively allosteric mechanisms that cause resistance to MTX, which is supported by a previous study into the allosteric regulation of DHFR. These results hint at more complex internal protein dynamics for function and resistance within DHFRs than previously estimated [67,78,85].

## Conclusion

Studying *P. jirovecii*'s DHFR and its evolution towards resistance to antifolates is of particular interest because of its insensitivity to classical antifungals. Here, we developed resources that will help study how resistance to antifolates evolves in this fungus. With the rising concerns about resistance in this pathogen, such resources will be useful to detect or predict resistance from DNA sequence alone, as well as potentially developing species-specific antifolates in the future. As other pathogenic fungi have been developing resistance to classical antifungals, it is more relevant than ever to study alternative treatment targets in fungal pathogens, such as antifolates [19].

## Supporting information

**S1 Fig. Gaussian mixture model optimization for MTX IC75 and IC90 conditions.** Optimization of information criterion for **a)** IC75 and **b)** IC90. Best Gaussian mixture model (dashed lines represent underlying Gaussians) to recapitulate the underlying distribution (black line/histogram) of **c)** IC75 and **d)** IC90. Density curve of significant mutants for Benjamini-Hochberg (green) correction is visible.
(TIFF)

**S2 Fig. PjDHFR protein expression analyses in *Saccharomyces cerevisiae* strain IGA130. a)** Construction schematics of tagged proteins expressed in yeast that can further be detected in protein cell extracts by western blots. All proteins tagged at their C-termini are tagged using 3xFLAG-tag, and all proteins tagged at their N-terminal are tagged using 1xFLAG-tag. N-terminal-tagged proteins were constructed with or without their native start codon. Constructions are not to scale. Schematic was made with BioRender. **b)** Western blots from biological replicates for all constructions. In C-terminal-tagged constructions with PjDHFR (wells 4-5-6), two bands can be seen, corresponding to the full length (top band) and the truncated protein from an alternative start codon (possibly M33 or M34, lower band). This lower band is absent from the wells with the N-terminal constructions (wells 7–8), supporting the hypothesis of an alternative start codon versus a post-translational cleavage.
(TIFF)

**S3 Fig. Dose response curve measurements for strain FDR0001 in the presence of MTX. a)** Growth curves of yeast strain FDR0001 (*dfr1Δ*) each expressing a different DHFR. Shaded areas represent confidence intervals across biological triplicates. The Empty vector does not

support growth in this strain. Strains with vector-expressed DHFRs show increased resistance, with PjDHFR being the most sensitive, followed by ScDHFR, and mDHFR(L22F/F31S) being fully resistant. **b)** Growth coefficient of strain FDR0001 with different DHFRs. $R^2$ for Hill equation fit: Empty = null, PjDHFR = 0.99, ScDHFR = 0.73 and mDHFR = >0.99.
(TIFF)

**S4 Fig. Correlation among biological replicates of competition assays in DMSO and MTX conditions. a)** Distributions of selection coefficients for amino acids in the different replicates in DMSO condition. Axes are scaled to allow comparison of the different conditions. Since the distribution of selection coefficients in DMSO is much narrower than in MTX, data appears as a group centered on 0. Density plots show the distribution of selection coefficients for the replicate on the x-axis. **b)** Distributions of selection coefficients for amino acids in the different replicates in MTX IC75 condition. **c)** Distributions of selection coefficients for amino acids in the different replicates in MTX IC90 condition. Statistical tests are Spearman's rank correlation.
(TIFF)

**S5 Fig. Effects of amino acid substitutions on the PjDHFR in DMSO and at IC75. a)** Selection coefficient of a given amino acid (y-axis) at a given position (x-axis) without MTX. Red points show the wild-type sequence of PjDHFR. **b)** Selection coefficient of a given amino acid (y-axis) at a given position (x-axis) at a MTX concentration corresponding to 75% of growth inhibition (IC75). **c)** Positions of contacting residues along PjDHFR modeled on structural alignments between PjDHFR and orthologous *P. carinii* DHFR (PDB: 3CD2 (MTX and NADPH) and 4CD2 (DHF)). Contact was established as amino acids with an α carbon located less than 8 Å from MTX, DHF or NADPH.
(TIFF)

**S6 Fig. Intersects between statistically significant mutants in IC75 and IC90.** Upset plot showing the intersections of the Benjamini Hochberg-FDR corrected groups (control for the false discovery rate of significantly resistant mutants at 5% confidence). The x-axis bars represent the size of each group and the y-axis represent intersection size for each group. For maximum confidence, mutations considered significant in both conditions should be considered.
(TIFF)

**S7 Fig. Individual growth curves for validation of mutations in DMSO and IC90.** Mutants are identified on top of their individual growth curves. For stop codons, we constructed both TAG (from DMS using NNK degenerate primers) and TAA as extra controls to ensure that TAG stop codons did not allow read through. All curves were done in triplicate, with error intervals between replicates appearing around the curve in a lighter color. Strains were grown in the same media as the competition assay for 40 hours, and growth rates were measured from these curves. Small discrepancies intervals in some of the curves in DMSO can be attributed to the formation of bubbles within the wells of the plates. These intervals were not considered when measuring growth rate.
(TIFF)

**S8 Fig. Comparison of growth rates from validation growth curves in strain FDR0001 for individually reconstructed mutants. a)** Growth rates in DMSO are plotted on the x-axis, as a measure of complementation, and growth rates in the presence of MTX IC90 are plotted on the y-axis, as a measure of resistance. These growth curves were conducted in the absence of β-estradiol. Many sensitive mutants cannot complement the deletion of *DFR1* in strain FDR0001 (*dfr1Δ*). Error bars represent growth rate values measured across the biological replicates in both conditions. Data points marked by crosses represent the Empty plasmid (orange), the wild-type

PjDHFR (blue), ScDHFR (yellow) and mDHFR (green). Other round data points are colored by their selection coefficients at IC90 unless specified otherwise. **b)** Same as panel a), but in the presence of β-estradiol, which leads to expression of MTX insensitive DfrB1. **c)** Comparison of growth rates between strains IGA130 (x-axis) and FDR0001 (y-axis) validation mutants in DMSO. All mutants can grow similarly in strain IGA130, but some mutants were non-functional in strain FDR0001. Color bar is the selection coefficient in DMSO. **d)** Comparison of growth rates between strains IGA130 (x-axis) and FDR0001 (y-axis) validation mutants in MTX IC90. For mutants that were identified as resistant, there is a strong correlation between growth rates measured in strain IGA130 and FDR0001, highlighting the minimal effect of the genomic *DFR1* on MTX resistance. Non-functional mutants in FDR0001 had to be supplemented with 100 nM β-estradiol to ensure growth of precultures. Statistical tests are Spearman's rank correlation. (TIFF)

**S9 Fig. Selection coefficients of stop codons in DMSO across positions.** Each point represents the selection coefficient of a stop codon at a given position (the median selection coefficient across triplicates). (TIFF)

**S10 Fig. Selection coefficients in DMSO and change in free energy of the protein (ddG) do not show the expected trend.** Using FoldX, changes in protein free energy were computed for all mutants forming a complex with DHF and compared to their selection coefficients in DMSO. Extreme values (-5<x<5 change in free energy) were scaled down to 5/-5, as extreme values of ddG measured by FoldX can often be attributed to clashes. Orange line represents the Lowess smoothed curve fitted using statsmodels default settings. Statistical tests are Spearman's rank correlation. (TIFF)

**S11 Fig. Comparison of changes in binding energy measured between PjDHFR F199 mutations with MTX and with DHF.** On the x-axis, we plot the change in binding energy between F199 mutants and DHF, and on the y-axis, between F199 mutants and MTX. The diagonal shows equal changes in binding energy to both molecules. Units are REU (Rosetta Energy Units). Statistical tests are Spearman's rank correlation. (TIFF)

**S12 Fig. Measurements of the effects of protein abundance on MTX resistance. a)** Flow cytometry measurements for different validation mutants and controls in strain IGA130. mEGFP represents mEGFP alone expressed under the regulation of the same promoter used for the screen. PjDHFR-mEGFP is the wild-type PjDHFR tagged with mEGFP. Mutants marked with sens or res are mutants that are identified as sensitive or resistant, respectively. Empty is the empty plasmid as a negative control. **b)** Correlation between median measured fluorescence and growth rate in MTX IC90, colored by mutant status. No correlation was found between expression levels and resistance. **c)** Anti-FLAG western blot and Ponceau red stains of strain FDR0003 (*DFR1* 3xFLAG-tag) to measure changes in expression levels of genomic *DFR1* when co-expressed with PjDHFR mutants. 11) is PjDHFR-mEGFP fusion protein, 12) PjDHFR is PjDHFR wild-type without fusion, 13) is mEGFP without fusion, and 14) is Empty plasmid. All transformations were done in strain FDR0003 unless specified. 15) Empty* is empty plasmid transformed in strain IGA130 as a FLAG-negative control. (TIFF)

**S1 Table. Strain information.** Table containing information for all strains used in this study. (XLSX)

**S2 Table. Plasmid information.** Table containing information for all plasmids used in this study.
(XLSX)

**S3 Table. Oligo information.** Table containing strain information for all oligos used in this study.
(XLSX)

**S4 Table. Media information.** Table containing strain information for all media used in this study.
(XLSX)

**S5 Table. PCR information.** Table containing strain information for all PCR reactions and cycles used in this study.
(XLSX)

**S6 Table. Validation mutants information.** Table containing strain information for all manually reconstructed mutants used in this study.
(XLSX)

**S7 Table. Mutations conferring resistance to some antifolates identified in literature.** Table containing mutations that were identified in previous studies as conferring resistance to some antifolates for some species of relevance in this study.
(XLSX)

**S1 File. Analysis of DMSO condition.** Demultiplexed and analyzed data for pooled competition assay in DMSO condition. Detailed readme is available in table Tab.
(CSV)

**S2 File. Analysis of MTX IC75 condition.** Demultiplexed and analyzed data for pooled competition assay in MTX IC75 condition. Detailed readme is available in table Tab.
(CSV)

**S3 File. Analysis of MTX IC90 condition.** Demultiplexed and analyzed data for pooled competition assay in MTX IC90 condition. Detailed readme is available in table Tab.
(CSV)

## Acknowledgments

We would like to thank the Landry lab for their helpful comments throughout the project. We also thank Joëlle Pelletier for her comments and insights.

## Author Contributions

**Conceptualization:** Francois D. Rouleau, Christian R. Landry.

**Data curation:** Francois D. Rouleau.

**Formal analysis:** Francois D. Rouleau, Soham Dibyachintan, Alicia Pageau.

**Funding acquisition:** Francois D. Rouleau, Christian R. Landry.

**Investigation:** Francois D. Rouleau, Alexandre K. Dubé, Isabelle Gagnon-Arsenault, Christian R. Landry.

**Methodology:** Francois D. Rouleau, Alexandre K. Dubé, Isabelle Gagnon-Arsenault, Soham Dibyachintan, Patrick Lagüe, Christian R. Landry.

**Project administration:** Francois D. Rouleau, Alexandre K. Dubé, Isabelle Gagnon-Arsenault, Christian R. Landry.

**Resources:** Francois D. Rouleau, Alexandre K. Dubé, Isabelle Gagnon-Arsenault, Patrick Lagüe, Christian R. Landry.

**Software:** Francois D. Rouleau, Soham Dibyachintan, Alicia Pageau, Philippe C. Després, Patrick Lagüe.

**Supervision:** Alexandre K. Dubé, Isabelle Gagnon-Arsenault, Christian R. Landry.

**Validation:** Francois D. Rouleau, Alexandre K. Dubé, Soham Dibyachintan, Patrick Lagüe, Christian R. Landry.

**Visualization:** Francois D. Rouleau.

**Writing – original draft:** Francois D. Rouleau.

**Writing – review & editing:** Francois D. Rouleau, Alexandre K. Dubé, Isabelle Gagnon-Arsenault, Soham Dibyachintan, Alicia Pageau, Philippe C. Després, Patrick Lagüe, Christian R. Landry.

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
