## [Decision Letter · Decision Letter 0]

4 Dec 2023

Dear Dr. Landry,

Thank you very much for submitting your Research Article entitled 'Deep mutational scanning of *Pneumocystis jirovecii* dihydrofolate reductase reveals allosteric mechanism of resistance to an antifolate' to PLOS Genetics.

You manuscript has been reviewed by 3 experts in microbial and protein evolution, and experts on DHFR. Overall, there is a consensus in the comments, both regarding impact and revisions suggested. Specifically, the following points need to be addressed:

1) Clearer explanation of the validity and caveats of the selection system for the deep mutational scan, particularly the use of MTX vs. TMP. See particularly comments of Reviewer 3.

2) Discussion on the use of MTX for PjDHFR in the clinic, if there is, to broaden the impact of the work.

3) Two reviewers cited alternative explanations for the observed mutations (e.g., position F199) than the one provided by the authors. The authors should provide further support for their argument or acknowledge the alternative explanation.

If you decide to revise the manuscript for further consideration at PLOS Genetics, please aim to resubmit within the next 60 days, unless it will take extra time to address the concerns of the reviewers, in which case we would appreciate an expected resubmission date by email to plosgenetics@plos.org.

We are sorry that we cannot be more positive about your manuscript at this stage. Please do not hesitate to contact us if you have any concerns or questions.

Yours sincerely,

Adrian Serohijos, PhD

Guest Editor

PLOS Genetics

Geraldine Butler

Section Editor

PLOS Genetics

---

Reviewer's Responses to Questions

**Comments to the Authors:**

Reviewer #1: Rouleau et al present a thorough study of methotrexate resistance mutations in Pneumocystis jirovecii DHFR, a potential drug target in a hard-to-study fungal pathogen. They use deep mutational scanning to comprehensively screen for resistance-causing mutations in two drug concentrations, and identify 355 mutations at 31 positions associated with resistance. One interesting finding is several seemingly “allosteric” mutations which result in resistance though they are located away from the active site. The paper is well written and exceedingly clear. It presents new tools for studying resistance in an important pathogen, and will be of strong relevance to geneticists and protein biochemists with interests in drug resistance. I was impressed with the rigor of the statistical analysis and the organization of the associated github repository, which together make the data processing transparent. In some respects the functional assay design seems suboptimal (which somewhat diminishes by enthusiasm for the work, see comments below), but the results presented are on the whole intriguing and well supported by the data.

Major comments:

1. It was not clear to me from the introduction if: (A) methotrexate is a common clinical treatment of P. jirovecii, with resistance an emerging threat, (B) trimethoprim (which would surely be less toxic) is the more commonly used clinical treatment, but didn’t work in their screen, hence they pivoted to MTX as another antifolate, or (C) antifolates as a whole aren’t usually used in P jirovecii treatment at present, but are under consideration so it is important to know how easy it is to evolve resistance. Which is the case? More explicit clarification on the connection between this work and current clinical practice (and potential future applications/impact of this assay – what would/could you screen next? Would/could you use this assay in drug design? ) would help in gauging the impact of this work.

2. A central contribution of this paper is “an experimental system and a resource for the study of resistance mutations to antifolates in the DHFR of P. jirovecii (PjDHFR)”. However, there are several places where I wondered about the efficacy of the assay as a general tool. In particular:

(A) The authors mention that S. cerevisiae is insensitive to TMP through mechanisms not involving DHFR. Could this be a problem for other antifolates or classes of antifolates?

(B) The screen was done in the background of the endogenous S. cerevisiae DFR1 because the deletion strain had poor transformation efficiency. As a consequence, the current assay does not present a way to assess the impact of mutations on function in the absence of drug. For example, one can’t tell from this assay if mutations at F31 show an activity-resistance tradeoff like the analogous mutations in other orthologs. Is there some way to address this in future work, or gauge this from existing data?

Overall it would help if the authors could include some discussion of assay limitations and potential ways to overcome them in future work.

3. I really like the idea that mutations at I10, I42, and F199 are acting allosterically to impact MTX binding. But could it also be possible that these mutations are instead enhancing stability and increasing pjDHFR abundance overall inside the cell?

Minor comments:

1. There seems to be a typo in equation 2, I believe the denominator in the second log () term should be median freq silent T_0 NOT T_final

2. Fig S2b, legend – line 1139, I think they mean alternative START codon, not STOP codon?

Reviewer #2: In the manuscript by Rouleau et al. the authors perform saturating mutational studies on P. jirovecii’s DHFR in yeast with the antifolate inhibitor and potent chemotherapeutic methotrexate (MTX). They chose methotrexate over trimethoprim (TMP) (an anti-infective DHFR inhibitor) as “S. cerevisiae has been reported as being insensitive to TMP through mechanisms not directly involving the DHFR”. However TMP is ~40nM inhibitor where MTX appears to be micromolar. While this maybe what was experimentally tractable in their study and appears to be a carefully performed experiments, the impact is lessened significantly as therapeutic relevance against MTX is not justified or explained.

In fact, a recent study Antimicrob Agents Chemother. 2022 Dec; 66(12): e00990-22. doi: 10.1128/aac.00990-22, PMCID: PMC9765006, PMID: 36317930 has shown that patients undergoing therapy with MTX do not get relief from P. jirovecii infections and resistance is not selected. In addition the authors do not compare their results with the clinical resistant mutations observed and tested on P. jirovecii’s DHFR - Antimicrob Agents Chemother. 2013 Oct;57(10):4990-8. doi: 10.1128/AAC.01161-13.Epub 2013 Jul 29, nor the structural comparisons with active and remote mutations J Chem Inf Model. 2021 Jun 28;61(6):2537-2541. doi: 10.1021/acs.jcim.1c00403. Epub 2021 Jun 17.

The structural figure 4 does not show sufficient detail to understand how the resistance mutations even within the active site may be conferring resistance to MTX as opposed to the mutational scanning for function. At quick glance the impact of mutations at F199 may be as simple as a repacking of the helix which packs against MTX.

Reviewer #3: In this manuscript, titled "Deep mutational scanning of Pneumocystis jirovecii dihydrofolate reductase reveals allosteric mechanism of resistance to an antifolate " by Rouleau FD, et. al., the authors explore methotrexate (MTX) resistance landscape of PjDHFR using deep mutational scanning (DMS) approach and pooled competition assay in yeast (S. cerevisiae). The authors derive selection coefficients for each substitution along the protein sequence by normalizing its log2-fold change in frequency between initial and final stages of growth in the absence and presence of two inhibitory concentrations of MTX (IC75 and IC90). Overall, 31 resistance-conferring positions along the protein sequence were discovered, for a total of 355 high-confidence resistance mutations. The authors then tested the validity of the obtained selection coefficients by engineering a fraction (44) of individual substitutions and testing their individual resistance effects. The authors concluded that most of the resistance-conferring positions along PjDHFR sequence identified in their study are located in close vicinity to the active site and coincide with the previously reported mutations in DHFRs of other origins. The analog of one antifolate-conferring substitution at position F199, however, was found only in bacterial and not eukaryotic DHFR sequences. Since F at position 199 is located far away from the active site and is buried within the hydrophobic core, the authors suggest an allosteric mechanism for substitutions at this position.

Major points.

1. The experimental system.

From the very beginning, the authors describe the development of a functional complementation assay to test the MTX resistance effects of mutations in PjDHFR. Specifically, in the Abstract, it is stated that " …by using functional complementation of the baker’s yeast ortholog…"; in the Introduction, "… We constructed yeast strains and plasmids that allow for the complementation of the Saccharomyces cerevisiae DHFR with PjDHFR". In the very first section of the Results, the authors introduce the design of the complementation system that is based on replacement of the yeast DHR1 gene with an MTX-resistance ortholog expressed from an inducible promoter and present the cartoon of the design as the first figure (Fig. 1a) in the manuscript. All this build up generates the ineluctable expectation that the presented functional complementation system will serve the basis for the mutational analysis. Particularly, because such a system could be used to test the effect of trimethoprim, the clinically relevant drug for P. jirovecii-caused pneumonia. However, this is not the case. The authors declare in as-a-matter-of-fact way that the system is ill suited for the high frequency transformation and, instead, a comprehensive scan of the mutational effects in PjDHFR is done on the background of functional DFR1 gene. This is a very misleading and confusing way of telling a story and should be avoided.

2. Validation of the pooled competition experiment in FDR0001 (dfr1-) strain.

Out of the 44 individually engineered validation mutants only a fraction was tested in the FDR0001 strain. The identity of these mutants and why these particular mutants were chosen for the analysis is not specified anywhere. It is also not clear how their effects on growth and MTX resistance are correlated with those in the IGA130 strain (WT for FDR1). The comparison of the resistance trends between the strains is also not possible, since Fig. 3a that depicts behavior in the IGA30 strain presents correlations between selection coefficients at IC75 and IC90 MTX concentrations, whereas Fig S7a that relates to FDR0001 strain shows correlation between growth rates in DMSO and IC90 MTX. But it does look that some of the mutants with high selection coefficient (red dots in Fig S7a) grew poorly in the presence of IC90 MTX in the FDR0001, suggesting that there is a potential discrepancy between the mutational effects in IGA130 and FDR0001 strains. Since MTX is a competitive inhibitor of enzyme activity, presence of the background DHFR can mask the resistance profile of the identified mutations, particularly because addition of MTX can upregulate the expression of the endogenous DHFR. Therefore, it is important to validate the mutational effects in the absence of the background endogenous expression in the FDR0001 strain using all 44 engineered mutants and compare the obtained effect with those in IG130 strain.

3. The allosteric mechanism of resistance

The correlation in the drop of the predicted binding energy of the MTX-PjDHFR complex with the selection coefficients of mutants in position 199 is not statistically significant, and the trend is driven entirely by the two substitutions towards charged residues. But even if the correlation was statistically significant, I do not think that it is sufficient to substantiate the claim of allosteric mechanism of resistance, particularly emphasizing it in the Title as the major insight from the work. It is true that position F199 is not participating in binding to DHF/MTX and NADPH. But this position is deeply buried in the hydrophobic core. Thus, the most trivial explanation for the drop in MTX-PjDHFR complex binding energy is the destabilization of PjDHFR, which effectively lowers the frequency of productive encounters with MTX (but less so with DHF). In that case, the effect of substitutions at this position is not allosteric in the true sense of the word because there is no structurally-based interaction between the distant locations – active site and position 199. Another possibility for the MTX resistance found for the mutants in this position is their effect on DHFR abundance. If these mutations increase the DHFR abundance, e.g., through dimerization o

---

## [Decision Letter · Decision Letter 1]

8 Apr 2024

Dear Dr Rouleau,

We are pleased to inform you that your manuscript entitled "Deep mutational scanning of *Pneumocystis jirovecii* dihydrofolate reductase reveals allosteric mechanism of resistance to an antifolate" has been editorially accepted for publication in PLOS Genetics. Congratulations!

Yours sincerely,

Adrian Serohijos, PhD

Guest Editor

PLOS Genetics

Geraldine Butler

Section Editor

PLOS Genetics

Comments from the reviewers (if applicable):

Dear Dr. Landry,

Two of the reviewers are satisfied with your revisions and have suggested to accept the manuscript. We still have not received comments from the remaining reviewer. But to avoid further delays and since all comments have been addressed, we have decided to accept your manuscript for publication.

Sincerely

Reviewer's Responses to Questions

**Comments to the Authors:**

Reviewer #1: The authors' revisions have satisfied my concerns. I especially appreciate that they took the time to measure protein expression/abundance levels for several key variants.

Reviewer #3: The authors have addressed all my concerns

**Have all data underlying the figures and results presented in the manuscript been provided?**

Reviewer #1: Yes

Reviewer #3: Yes

PLOS authors have the option to publish the peer review history of their article (what does this mean?). If published, this will include your full peer review and any attached files.

Reviewer #1: No

Reviewer #3: No

**Data Deposition**

http://datadryad.org/submit?journalID=pgenetics&manu=PGENETICS-D-23-01160R1

**Press Queries**

---

## [Editor Report · Acceptance letter]

23 Apr 2024

PGENETICS-D-23-01160R1 

Deep mutational scanning of *Pneumocystis jirovecii* dihydrofolate reductase reveals allosteric mechanism of resistance to an antifolate 

Dear Dr Rouleau, 

We are pleased to inform you that your manuscript entitled "Deep mutational scanning of *Pneumocystis jirovecii* dihydrofolate reductase reveals allosteric mechanism of resistance to an antifolate" has been formally accepted for publication in PLOS Genetics! Your manuscript is now with our production department and you will be notified of the publication date in due course.

With kind regards,

Anita Estes

PLOS Genetics

On behalf of:
